# IMAGE GENERATION FROM CONTEXTUALLY CONTRA-DICTORY PROMPTS

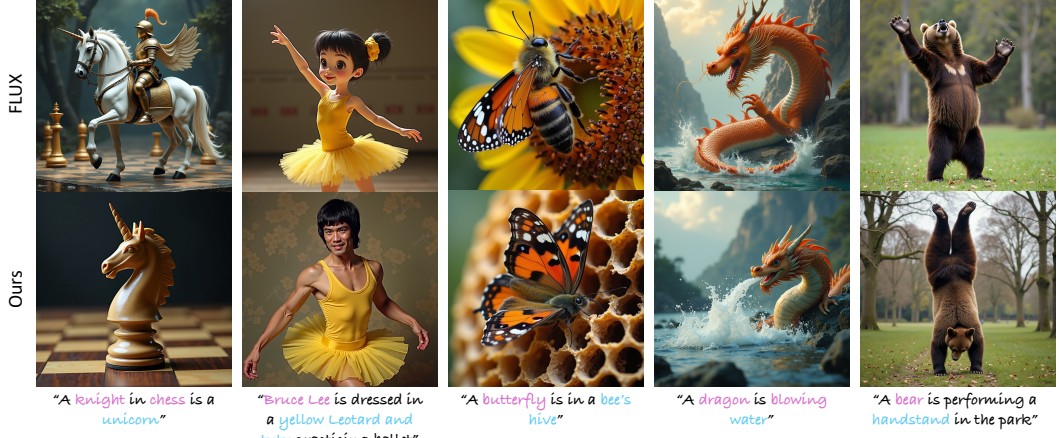

Figure 1: *Our method addresses contextual contradictions in text-to-image generation. These contradictions arise when one concept implicitly conflicts with another due to the model's learned associations. For example, if a concept like "butterfly" is strongly entangled with "flowers", it may conflict with another concept in the prompt such as "bee's hive", leading the model to ignore or distort part of the semantic meaning.*

## ABSTRACT

Text-to-image diffusion models excel at generating high-quality, diverse images from natural language prompts. However, they often fail to produce semantically accurate results when the prompt contains concept combinations that contradict their learned priors. We define this failure mode as *contextual contradiction*, where one concept implicitly negates another due to entangled associations learned during training. To address this, we propose a stage-aware prompt decomposition framework that guides the denoising process using a sequence of proxy prompts. Each proxy prompt is constructed to match the semantic content expected to emerge at a specific stage of denoising, while ensuring contextual coherence. To construct these proxy prompts, we leverage a large language model (LLM) to analyze the target prompt, identify contradictions, and generate alternative expressions that preserve the original intent while resolving contextual conflicts. By aligning prompt information with the denoising progression, our method enables fine-grained semantic control and accurate image generation in the presence of contextual contradictions. Experiments across a variety of challenging prompts show substantial improvements in alignment to the textual prompt.

## 1 INTRODUCTION

Text-to-image diffusion models have demonstrated remarkable capabilities in generating high-quality and diverse visual content from natural language prompts (Rombach et al., 2022; Ramesh et al., 2021; Ho et al., 2020). However, achieving precise semantic alignment between the generated image and the conditioning prompt remains an open challenge.

This issue becomes particularly pronounced when the target prompt lies outside the model's training distribution, such as prompts that combine semantically plausible yet statistically uncommon or unconventional concepts. For example, as shown in Figure 1, generating an image from the prompt *"A butterfly is in a bee's hive"* often results in a butterfly on a flower. This is due to the model's prior that entangles butterflies with flowers, which implicitly contradicts the notion of a bee's hive.

We refer to this phenomenon as *Contextual Contradiction*, a conflict between two concepts that arises not from direct semantic opposition, but from the model's associations learned during training. More precisely, we say that concept $A$ contextually contradicts concept $B$ if the model's prior entangles $A$ with concept $C$, and $B$ contradicts $C$ (see Figure 2). In Figure 1, we illustrate this with a blowing dragon, which stands in contextual contradiction with the water due to its entanglement with fire.

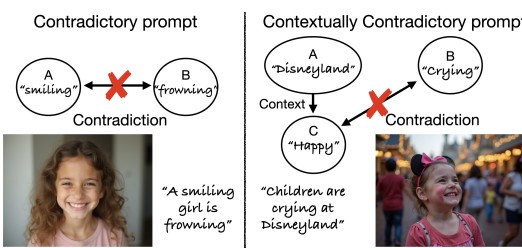

Figure 2: *On the left is a direct contradiction, since a girl cannot smile and frown at the same time. On the right is a contextual contradiction: while Disneyland and crying are not directly opposed, the model's prior associates Disneyland with happiness, which conflicts with crying.*

The phenomenon of contextual contradiction in text-to-image models relates to the broader issue of spurious correlations in deep learning. Models often exploit shortcuts, relying on correlations in the training data that are statistically strong but semantically misleading (Geirhos et al., 2020). In this paper, we identify a similar bias in generative models: contextual contradictions occur when prompts combine concepts that individually align with the model's priors but conflict when combined, revealing the model's reliance on such correlations rather than robust compositional reasoning.

To address this issue, we introduce *Stage-Aware Prompting (SAP)*, which builds on the observation that the denoising process follows a coarse-to-fine progression, during which different semantic attributes (e.g., background, pose, shape, and texture) emerge at distinct stages (Chefer et al., 2023; Balaji et al., 2022; Patashnik et al., 2023). Our key idea is to guide the model at each stage of denoising with the information most relevant to the type of content being formed at that point. To achieve this, we decompose the original prompt into a sequence of proxy prompts, each aligned with the attributes expected at a specific stage and designed to avoid contextual contradictions.

Ensuring that proxy prompts preserve the original intent while avoiding contextual contradictions requires a broad understanding of the real world. For example, it involves understanding that a bear is entangled with specific poses, such as walking on all fours or standing upright, which in turn, contradicts the handstand pose. To achieve this, SAP leverages a large language model (LLM) to analyze the target prompt, identify contextual contradictions, and construct suitable proxy prompts.

We demonstrate that, by using in-context examples and prompting the LLM to follow a reasoning process through a brief explanation, it can identify contextually contradictory concepts in a prompt and determine the appropriate stage of denoising at which each attribute should be introduced. It can also suggest alternative, non-conflicting concepts that preserve the intended attributes and use them to construct stage-specific proxy prompts. In doing so, the LLM effectively guides the model toward the intended meaning of the original prompt.

Through extensive experiments, we demonstrate the effectiveness of SAP in generating images from contextually contradictory text prompts. By introducing prompt information at targeted stages, SAP generates precise combinations of semantic attributes while avoiding undesired entanglement. Compared to previous methods, SAP's stage-dependent prompt decomposition leads to more faithful and semantically aligned generations.

## 2 RELATED WORK

**Learned Spurious Correlations** Machine learning models are known to be sensitive to spurious correlations in their training data (Geirhos et al., 2020; McCoy et al., 2019; Ye et al., 2024), leading to performance drops when training-time associations do not hold at test time. Prior work has extensively studied this in discriminative vision tasks, showing, for example, that recognition models tend to rely heavily on background cues (Singh et al., 2020; Xiao et al., 2021; Beery et al., 2018).

In our work, we show that text-to-image models exhibit similar behavior. When given contextually contradictory prompts – combinations of concepts that conflict with correlations seen during training – diffusion models often fail to generate images that accurately reflect the prompt. We evaluate this behavior using the Whoops! dataset (Bitton-Guetta et al., 2023), which contains prompts constructed by first describing two co-occurring elements, and then replacing one with a less compatible alternative. This results in scenarios that are unlikely to occur in the real world.

**Semantic Alignment in Text-to-Image Synthesis**    Text-to-image models often struggle to fully capture the semantic intent of input prompts, particularly when prompts involve complex or internally conflicting concepts. Previous works have analyzed common failure cases and proposed targeted improvements across various stages of the generation pipeline, including enhanced text embedding representations (Rassin et al., 2022; Feng et al., 2022; Tunanyan et al., 2023), refined attention mechanisms Dahary et al. (2024), guidance strategies that leverage attention maps for loss heuristics (Chefer et al., 2023; Rassin et al., 2023; Meral et al., 2024; Agarwal et al., 2023) and dynamic guidance scheduling via annealed classifier-free guidance (Yehezkel et al., 2025). Despite these advances, existing methods often fail to handle prompts containing contradictory concepts arising from the model's learned associations. Our work directly addresses this underexplored challenge, focusing on contextually contradictory prompts.

**Multi-Prompt Conditioning Techniques**    Conditioning diffusion models on multiple prompts has emerged as an effective strategy for improving control and compositionality. One line of work, primarily focused on personalization, introduces distinct learned tokens at different layers of the model and at various denoising timesteps (Alaluf et al., 2023; Voynov et al., 2023). This design allows each token to capture different attributes of the personalized concept, leading to improved identity preservation. Other approaches vary the prompt across timesteps to modulate specific visual properties, such as object shape (Liew et al., 2022; Patashnik et al., 2023), or alternate between rare and frequent object descriptions to improve attribute binding (Park et al., 2024). Fine-grained spatial control has also been achieved by assigning sub-prompts to separate image regions (Yang et al., 2024). Additionally, some methods leverage multiple diffusion models, each conditioned on different prompt attributes, and combine their outputs into a unified prediction (Liu et al., 2022; Bar-Tal et al., 2023). Unlike most prior works, we focus on utilizing multiple prompts to settle internal semantic tension, where concept combinations lead to contextual contradictions.

**LLM-Guided Diffusion**    LLMs have demonstrated strong capabilities in language understanding. They also capture broad world knowledge through large-scale training on diverse text. Recent approaches have leveraged these capabilities to guide diffusion model generation, often incorporating planning and reasoning to improve semantic alignment (Yang et al., 2024; Park et al., 2024; Hu et al., 2024). In our work, we employ LLMs with in-context learning to identify contextual contradictions, generate proxy prompts, and determine the corresponding timestep ranges for conditioning, while encouraging reasoning through brief explanatory outputs.

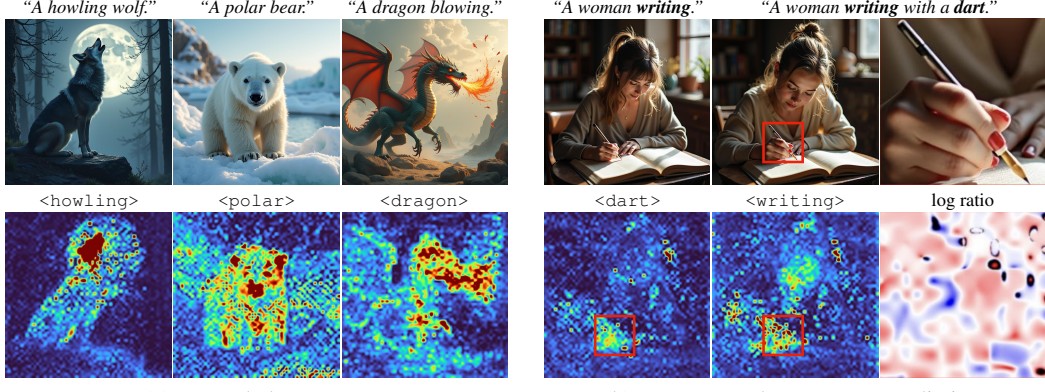

(a) Entangled concepts.    (b) From entanglement to contradiction.

Figure 3: *(a) By examining attention maps, we observe that textual tokens embed contextual associations, leading to the generation of concepts not explicitly mentioned in the prompt. For example, the token 'howling' encourages the presence of the moon, as indicated by its strong attention connection. (b) In prompts with contextual contradictions, two tokens may overlap in attending to the same region, as seen with 'dart' and 'writing'. The token 'writing' dominates this region, as shown in the log-ratio map, where red areas indicate stronger attention to 'writing' relative to 'dart'.*

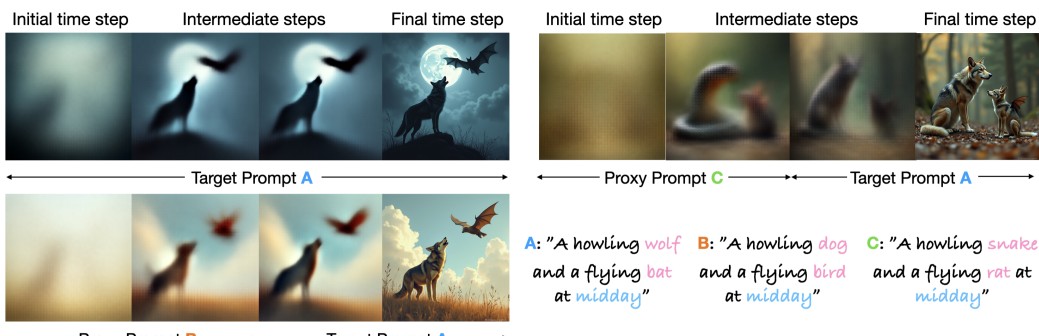

Figure 4: *Coarse-to-fine denoising with stage-aware prompting. We show $x_0$ predictions at initial, intermediate, and final steps. A is the target prompt, B a suitable proxy, and C an unsuitable proxy. Using A alone locks in night and moon despite "midday". Using B first and then switching to A preserves a daytime layout and later adapts the identities to wolf and bat. Using C first sets a layout without a flying object, so the final image fails to produce the intended subjects.*

## 3 FROM ENTANGLED CONCEPTS TO CONTEXTUAL CONTRADICTIONS

Diffusion models inherit strong distributional biases from their training data, where objects are frequently tied to specific contexts. For example, prompts like *"a duck"* almost always result in water backgrounds, and *"a polar bear"* appears in snow. These reflect *entangled concepts*, learned associations that go beyond the explicit text. While often helpful, such priors hinder generation when prompts require unusual or contradictory combinations.

To study this effect, we analyze attention maps (Figure 3a), which indicate the spatial regions influenced (attended) by each token. We find that text tokens often attend not only to the image regions directly corresponding to the object but also to contextually linked elements. For example, in *"a howling wolf"*, the token 'howling' influences both the mouth and the moon. Similarly, 'dragon' attends to flames even when fire is not mentioned. These patterns reveal that the model encodes distributional correlations beyond literal semantics.

These distributional correlations contribute to the difficulty of generating images with *contextual contradictions*. For example, when generating an image from the prompt *"a woman writing with a dart"*, the model fails to replace the pen with a dart. The attention maps shed light on this failure: both 'writing' and 'dart' attend to the same spatial area (the hand/tool), but 'writing' dominates (see log-ratio maps in Figure 3b), suppressing the influence of 'dart'. This reflects a failure mode in which entrenched associations override less familiar ones, preventing proper integration of conflicting concepts.

## 4 STAGE AWARE PROMPTING (SAP)

In this section, we begin by analyzing the coarse-to-fine behavior of the denoising process (Section 4.1). Building on the insights from our analysis, we introduce our training-free framework for resolving contextual contradictions in text-to-image generation. As illustrated in Figure 5, our approach consists of two main components: (i) prompt decomposition (Section 4.2, top part of the figure), and (ii) stage-aware prompt injection (Section 4.3, bottom part of the figure). In the following, we describe each of these components.

### 4.1 COARSE-TO-FINE DENOISING

Diffusion models generate images in a coarse-to-fine manner: early steps determine low-frequency structures, while high-frequency details emerge in later steps (Hertz et al., 2022; Balaji et al., 2022; Chefer et al., 2023; Patashnik et al., 2023). From this behavior, we draw two key observations: (i) *Irreversibility of details.* As denoising progresses, the model sequentially commits to different levels of detail, beginning with layout and overall shape. Once these are formed, they cannot be revised in later stages, even if they conflict with the prompt. (ii) *Flexibility in high-frequency details.* In early stages, high-frequency details are absent and unaffected by the prompt, enabling flexible guidance without influence on fine details.

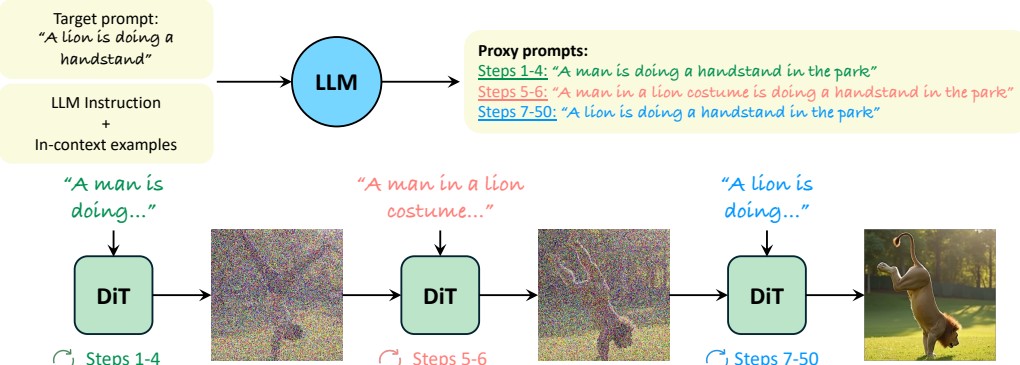

Figure 5: *SAP generates images from contextually contradictory prompts using time-dependent proxy prompts. (Top) A large language model (LLM) decomposes the target prompt into a sequence of proxy prompts with corresponding timestep intervals. (Bottom) These proxy prompts are injected into the diffusion process at their designated intervals to guide generation.*

As shown in Section 3, contextual contradictions stem from concept entanglement in the diffusion prior. Since different concepts emerge at different levels of detail during the coarse-to-fine denoising process, they can be decomposed across the denoising stages. We illustrate this in Figure 4 by examining the model's $x_0$ predictions across denoising steps. In the top-left row, the prompt *"a howling wolf and a flying bat at midday"* shows that early steps already impose entangled nighttime and moon structures, contradicting "midday". In the bottom-left row, starting with a *proxy prompt* containing "dog" and "bird" (instead of "wolf" and "bat", respectively) and later switching to the target prompt produces a correct midday scene with the intended objects. The bottom-left row demonstrates both the *irreversibility of coarse details* (the scene remains daytime) and the *flexibility of high-frequency details* (the object identities adapt). The top-right row highlights the importance of selecting an appropriate proxy prompt. Since rats do not fly, the layout determined in early steps lacks a flying object, resulting in a sitting wolf with bat wings.

These observations motivate a stage-aware prompting strategy that fixes structural decisions early while keeping later attributes flexible. Building on this, we now describe the two main components of our method: prompt decomposition and stage-aware prompt injection.

### 4.2 PROMPT DECOMPOSITION

Given a prompt $P$ that contains contextually contradictory concepts, we aim to construct a sequence of proxy prompts $\{p_1, p_2, ..., p_n\}$ and corresponding timestep intervals $\{I_1, I_2, ..., I_n\}$ that together reflect the intended semantics of $P$. Each proxy prompt $p_i$ is designed to (i) preserve the relevant semantics of $P$ for attributes typically formed during its interval $I_i$, which is crucial due to the *irreversibility of details*, and (ii) avoid contradictions likely to emerge at that stage, leveraging the *flexibility in high-frequency details*. This decomposition conditions the diffusion model on contextually coherent content that evolves in tandem with the coarse-to-fine denoising process.

To generate proxy prompts and their intervals, we use a large language model (LLM) that detects contextual contradictions and proposes suitable substitutes for conflicting concepts. It also infers the appropriate staging of these concepts within the proxy prompts. We implement this using a structured prompt template containing instructional text, in-context examples, and explanations of contextual contradictions. The examples demonstrate both successful decompositions and cases requiring no decomposition, enabling the LLM to generalize. The full instruction prompt is provided in the Appendix (see Table 7).

**In-context Examples.** Our in-context examples (Table 8) take a target prompt as input and output proxy prompts with timestep intervals, along with a brief explanation of the contradiction. Requiring the LLM to provide this explanation encourages reasoning about conflicts and ensures coherent substitutions. These examples were created by identifying contextually contradictory prompts that fail under the base model (FLUX) and manually decomposing them into proxy prompts with corresponding intervals. The explanations were auto-generated by an LLM. We include 20 examples that demonstrate diverse strategies for handling contradictions. We next elaborate on one of the most frequent strategies in our method.

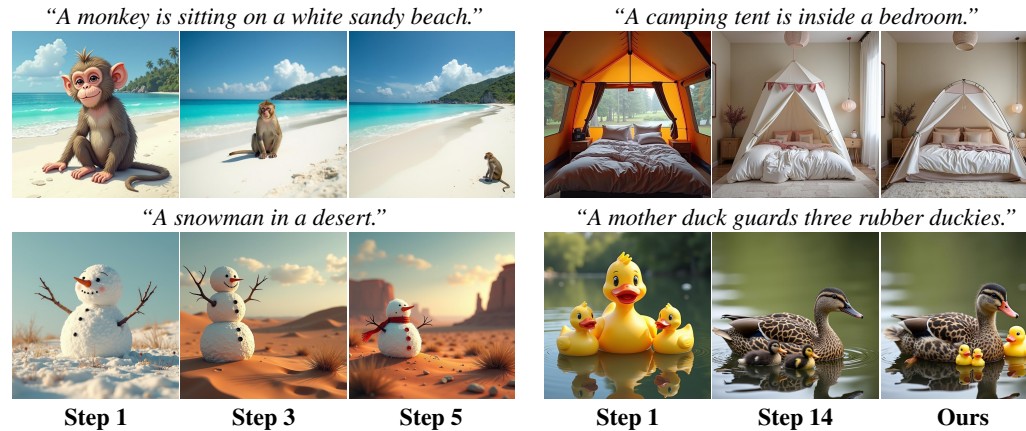

*"A monkey is sitting on a white sandy beach."*   *"A camping tent is inside a bedroom."*

*"A snowman in a desert."*   *"A mother duck guards three rubber duckies."*

**Step 1**   **Step 3**   **Step 5**   **Step 1**   **Step 14**   **Ours**

Figure 6: *Early insertion of the foreground allows the model to allocate more space to it, while late insertion confines the object within the existing layout, making it appear smaller (e.g., the snowman introduced at Step 5). Step labels indicate when the foreground object is introduced.*

Figure 7: *Effect of interval assignment. Introducing the full prompt too early fails to disentangle contextual contradictions, while introducing it too late alters only fine details. Top proxy: "A pillow fort in a bedroom"; bottom proxy "A mother duck guards three ducklings".*

**Concept Substitution.** In this strategy, a conflicting concept is temporarily replaced with a structurally appropriate proxy (Figure 4). A simpler alternative is to omit the conflicting concept, but introducing an object only in later stages without a placeholder can distort its perceived size or cause it to be omitted entirely. In Figure 6, we demonstrate this by comparing decompositions that differ only in when the second interval begins. The first proxy specifies the background, while the second adds the foreground. Introducing the foreground early allows the model to allocate more space, whereas delaying it constrains the layout and produces a smaller object. Substitution resolves these issues and yields stable layouts. In Figure 7, we show the effect of misplacing intervals across denoising stages. Using two proxies with different intervals, where the second is the full prompt, we observe two failure modes: introducing the full prompt too early prevents disentangling contradictions, while introducing it too late alters only fine details. Earlier in Figure 4 we illustrated the importance of careful proxy selection.

### 4.3 STAGE-AWARE PROMPT INJECTION

Given a sequence of proxy prompts $\{p_1, p_2, ..., p_n\}$, their corresponding timestep intervals $\{I_1, I_2, ..., I_n\}$, and a text-to-image (T2I) diffusion model, we condition the model using different prompts throughout the denoising process. At each timestep $t$, we apply the prompt $p_i$ such that $t \in I_i$. By aligning each proxy prompt with its interval, the denoising process is guided by concepts appropriate to the level of detail emerging at that stage. This enables gradual image construction while avoiding conflicts with the model's learned priors. The injection mechanism integrates seamlessly into existing inference pipelines without architectural modifications and is compatible with a range of pretrained diffusion models.

## 5 EXPERIMENTS AND RESULTS

In this section, we evaluate SAP through qualitative (Section 5.1) and quantitative (Section 5.2) comparisons. We further conduct ablation studies (Section 5.3) on component contributions and robustness, followed by a discussion on the limitations of our method.

**Implementation Details.** We use FLUX.1 [dev] (Labs, 2024) as the base T2I model and GPT-4o (Achiam et al., 2023) for prompt decomposition. In all experiments, inference is performed using $T = 50$ steps and the LLM is restricted to at most three proxies per prompt. Baseline hyperparameters follow their original papers or implementations, with $T = 50$ steps for fair comparison. To further demonstrate robustness, we also report results with SD3.0 using the same LLM-generated proxy prompts and intervals.

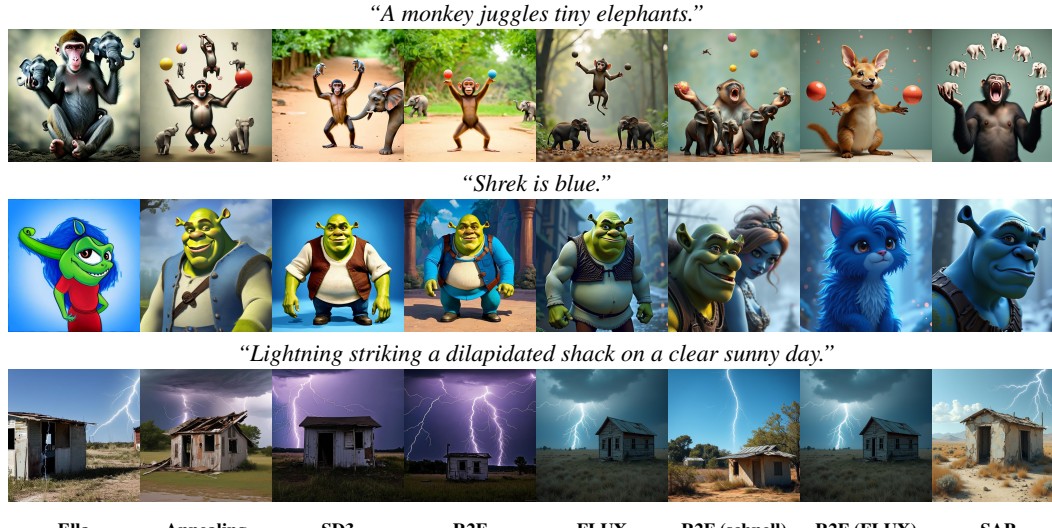

*"A monkey juggles tiny elephants."*

*"Shrek is blue."*

*"Lightning striking a dilapidated shack on a clear sunny day."*

| Ella | Annealing | SD3 | R2F | FLUX | R2F (schnell) | R2F (FLUX) | SAP |

Figure 8: *Qualitative comparison of SAP with baseline methods. Our method resolves contextual contradictions, whereas baselines struggle to produce text-aligned images. Additional examples are provided in the Appendix.*

**Baselines.** We compare SAP against the following approaches: (1) base models FLUX-dev (denoted by FLUX) (Labs, 2024) and SD3.0 (Esser et al., 2024); (2) R2F (Park et al., 2024), a training-free method reported under three settings: SD3.0 (original), FLUX-schnell (official), and our reimplementation on FLUX; (3) Annealing Guidance (Yehezkel et al., 2025), which trains a small MLP to predict the classifier-free guidance scale at each step; and (4) Ella (Hu et al., 2024), a fine-tuned model on SD1.5.

**Datasets.** We evaluate SAP using three datasets: Whoops! (Bitton-Guetta et al., 2023), Whoops-Hard, and ContraBench. *Whoops!* consists of 500 prompts paired with commonsense-defying images, designed to test visual reasoning and compositional understanding. While relevant to our task, many of its prompts are relatively easy for modern T2I models and do not consistently expose model limitations.

To address this, we curate *Whoops-Hard*, a subset of 100 particularly difficult prompts from Whoops!, providing a stronger benchmark for evaluating current state-of-the-art models. To further probe semantic alignment under contradictory conditions, we introduce *ContraBench*, a curated set of 40 prompts capturing contextual contradictions. The dataset was constructed in two steps: (1) ChatGPT generated candidate prompts based on the definition of contextual contradictions, and (2) human annotators filtered them to retain only those that clearly expressed contradictions. The full prompt lists for both datasets are provided in the Appendix (Tables 10 and 11).

## 5.1 QUALITATIVE RESULTS

Figures 8, 11 and 12 present qualitative comparisons on the Whoops! and ContraBench datasets. Across both benchmarks, baseline methods consistently exhibit characteristic failure modes when handling contradictory prompts. In contrast, SAP successfully generates challenging cases such as a blue Shrek or a monkey juggling tiny elephants (Figure 8). In addition to FLUX, SAP also improves SD3 generations under contradictory prompts (Figure 9).

For SD3 and FLUX, contradictory prompts expose conflicts with learned priors, resulting in prompt misalignment. Ella and Annealing Guidance, not designed for contradictions, perform less effectively on such cases. R2F alternates between prompts at predefined timesteps, a strategy designed for attribute binding rather than addressing contextual contradictions. While it can reinforce rare concepts, it does not align prompts with the stages at where semantic features emerge during denoising. As a result, it often produces hybrid concepts that merge incompatible elements from conflicting concept (see the bodybuilder in Figure 9 and the owl and SpongeBob in Figure 11).

In contrast, SAP produces semantically coherent outputs by introducing proxy prompts at denoising stages where corresponding features emerge. This enables effective handling of conflicting concepts.

Table 1: *Quantitative evaluation on various benchmarks using the GPT-4o vision-language model. We report average alignment, where alignment reflects how well the image matches the prompt semantics, independent of visual quality. SAP achieves the best results, regardless of the base model.*

| Models | Benchmarks | | |
| | Whoops | Whoops-Hard | Contra-Bench |
|---|---|---|---|
| SD3.0 | 82.63 | 55.73 | 57.5 |
| FLUX | 78.85 | 44.3 | 57.16 |
| Ella | 69.09 | 45.19 | 55.16 |
| Annealing | 79.59 | 59.06 | 58.33 |
| R2F | 83.50 | 57.06 | 59.16 |
| R2F$_{schnell}$ | 79.58 | 54.80 | 59.33 |
| R2F$_{FLUX}$ | 48.68 | 32.80 | 25.33 |
| SAP$_{SD3.0}$ | **85.87** | **64.40** | 65.33 |
| SAP | 85.10 | 62.13 | **66.16** |

Table 2: *User study results. Win rates of SAP in text-image alignment and image quality, compared against each baseline method.*

| | SD3 | FLUX | Ella | Annealing | R2F |
|---|---|---|---|---|---|
| Alignment | 70% | 81% | 81% | 73% | 75% |
| Quality | 72% | 63% | 74% | 79% | 68% |

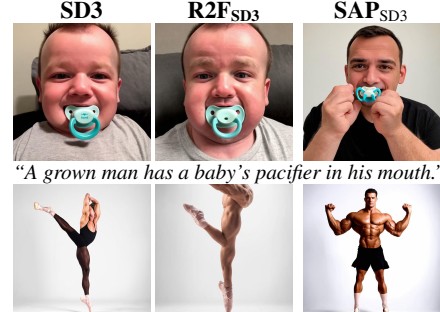

*"A grown man has a baby's pacifier in his mouth."*

*"A bodybuilder balancing on pointe shoes."*

Figure 9: *SAP is robust to the base model, as shown by the results obtained with SAP$_{SD3}$.*

Across both benchmarks, SAP consistently generates coherent images for contradictory prompts. Robustness is further demonstrated in Figure 13, comparing generations across multiple seeds.

## 5.2 QUANTITATIVE RESULTS

We evaluate prompt alignment using GPT-4o vision–language model (VLM). For each generated image, GPT-4o assigns a score from 1–5 based on adherence to the prompt. Scores are averaged across three fixed random seeds per prompt and scaled to 20–100. The evaluation prompt is provided in the Appendix (Table 12).

As shown in Table 1, SAP outperforms all baselines across the three benchmarks. Between the base models, SD3.0 tends to yield stronger alignment under contradictions, while FLUX offers higher visual quality (Figure 8). SAP improves both backbones, enhancing prompt adherence while maintaining the visual fidelity of the underlying models (Figures 8 and 9 ).

**User study.** VLM-based metrics often miss subtle semantic inconsistencies and do not adequately assess image quality. To complement them, we conducted a user study evaluating both prompt adherence and overall visual appearance. We randomly sampled 24 prompts from the Whoops! and ContraBench benchmarks. For each prompt, participants compared two images, one generated by SAP and the other by a baseline, and answered: (1) which most accurately reflected the prompt, and (2) which had higher visual quality. In total, we collected responses from 61 users, yielding 1,464 individual evaluations. Table 2 summarizes the win rates of SAP against each baseline.

These results highlight the superiority of our approach in handling contextually contradictory prompts, achieving both stronger prompt alignment and higher visual quality.

## 5.3 ABLATION STUDIES

We evaluate SAP through ablations that assess design choices in prompt decomposition and robustness under different conditions, including interval perturbations and alternative LLMs. Additional results on non-contradictory prompts are in the Appendix.

Table 3: *Ablation on Whoops-Hard. We evaluate our prompt decomposition method by (1) removing in-context examples, (2) removing the explanation requirement, and (3) limiting decomposition to two proxy prompts.*

| | static | w/o in-context | w/o reasoning | 2 proxy | Full |
|---|---|---|---|---|---|
| Alignment | 44.3 | 48.0 | 46.46 | 60.26 | 62.13 |

**Prompt decomposition components.** We conduct ablations on the Whoops-Hard benchmark, where each variant isolates a design choice to quantify its effect on alignment within our method

Table 4: *Effect of perturbing LLM-predicted timestep intervals. Boundaries are uniformly shifted within window w. $SAP_{w=i}$ denotes evaluation with window i.*

| | Benchmarks | | |
|---|---|---|---|
| **Models** | **Whoops** | **Whoops-Hard** | **Contra-Bench** |
| FLUX | 78.85 | 44.3 | 57.16 |
| SAP | 85.10 | 62.13 | 66.16 |
| $SAP_{w=3}$ | 84.18 | 62.06 | 65.5 |
| $SAP_{w=5}$ | 81.46 | 58.46 | 62.5 |

Table 5: *Performance of SAP when combined with different LLMs, comparing GPT-4o and Llama-3.1-8B-Instruct.*

| | Benchmarks | | |
|---|---|---|---|
| **Models** | **Whoops** | **Whoops-Hard** | **Contra-Bench** |
| FLUX | 78.85 | 44.3 | 57.16 |
| $SAP_{GPT4o}$ | 85.10 | 62.13 | 66.16 |
| $SAP_{Llama3.1}$ | 80.52 | 59.53 | 61.16 |

(Table 3). In-context examples significantly improve the LLM's ability to decompose contradictory prompts, leading to better text–image alignment. Removing the explanation requirement impairs reasoning and causes a notable drop, showing that generating explicit explanations encourages more coherent semantic decisions. Restricting decomposition to two proxies performs close to the full method, while allowing up to three proxies provides extra flexibility for harder cases and yields further gains.

**Robustness to LLM-predicted timestep intervals.** Our method relies on LLM-predicted intervals to schedule proxy prompts, but these boundaries do not require exact placement. The earlier results (Figure 7) highlight that placing proxy prompts at the wrong *stage* of denoising (e.g., too early or too late) can harm alignment. Here we show that within the correct coarse stage, the method is robust to moderate boundary shifts. Specifically, we perturb interval boundaries while keeping the proxy prompts fixed, uniformly shifting them within windows of varying size. As shown in Table 4, small shifts of up to $\pm 1$ step (window=3) have almost no effect on alignment, and even larger shifts of up to $\pm 2$ steps (window=5) cause only minor degradation, despite representing a substantial perturbation relative to the full method effective range ($\sim$12 steps; see Table 7). These results confirm that SAP is sensitive to the stage at which information is introduced, but largely insensitive to exact step boundaries within that stage.

**Robustness across LLMs.** Since our framework hinges on LLM-driven prompt decomposition, we further examined its robustness under different language models. We evaluated both a proprietary model (GPT-4o) and a comparatively lightweight open-source alternative (LLaMA-3.1-8B-Instruct). While GPT-4o delivers the strongest performance, the smaller LLaMA-3.1-8B-Instruct still yields consistent improvements over the baseline (see Table 5).

**Limitations.** In the figure to the right, we present three failure cases of our method on examples from the Whoops! benchmark. Since our approach relies on guiding the model through text alone, it cannot control properties that the underlying model inherently struggles with, such as generating specific quantities or enforcing precise orientations.

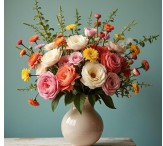 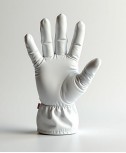 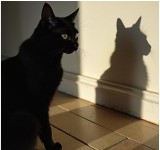

"A bouquet of flowers is upside down in a vase"  "A white glove has 6 fingers"  "The shadow of a cat is facing the opposite direction"

## 6 CONCLUSIONS

We introduced a training-free framework for resolving contextual contradictions in text-to-image generation, cases where seemingly plausible prompts fail due to strong, hidden model biases. At its core, our method leverages the coarse-to-fine generation process to separate contradictions across denoising stages, enabling faithful rendering of prompts that would otherwise yield semantically inconsistent outputs. The introduction of proxy prompts steers the generative process in line with the model's priors, enabling it to resolve conflicts and preserve semantic fidelity without the need for retraining or fine-tuning.

We argue that since our approach already leverages the broad world knowledge of vision–language models, integrating them more tightly with generative models holds promise for addressing contextual contradictions directly. As a next step, we plan to explore emerging compound architectures that combine VLMs and generative models, with the aim of understanding how to effectively harness them to resolve conflicts in open-ended generation.

# 7 ETHICS STATEMENT

Our work contributes to improving the semantic alignment of text-to-image models under contradictory or biased prompts. As a consequence, our method enhances users' ability to control generative models and faithfully render contradictory concepts. While this provides positive benefits, such as reducing unintended biases and enabling more inclusive image generation, it also increases the potential for misuse, including the creation of harmful, misleading, or inappropriate content. As with any advance in generative modeling, these dual-use concerns highlight the importance of responsible deployment, safeguards, and continued ethical oversight to ensure that such improvements contribute positively to society.

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

## A ADDITIONAL RESULTS

**Improved realism.** SAP generates photorealistic and semantically coherent images for prompts with atypical attribute combinations (Figure 10). In contrast, FLUX often defaults to cartoon-like renderings, even when photorealism is explicitly requested, revealing a contextual contradiction between fantastical content and realistic style. By using contradiction-free proxy prompts, SAP avoids these biases and produces realistic outputs regardless of whether photorealism is explicitly required in the prompt.

**Non-contradictory prompts.** To ensure applicability in general text-to-image scenarios, we verify that our method does not negatively affect prompts without contextual contradictions. We find that including even a single non-contradictory in-context example is sufficient for the LLM to default to using the full prompt in such cases. We evaluate this behavior using GPT-4o alignment scores on the PartiPrompts-Simple benchmark, which contains simple, non-contradictory prompts (Table 6).

**Additional qualitative comparisons.** Figures 11 and 12 present additional qualitative comparisons of our method, while Figure 13 shows results across multiple seeds.

Table 6: *Alignment performance on the PartiPrompts-simple benchmark, which contains simple, non-contradictory prompts. Scores are computed using GPT-4o vision-language model. Our method achieves comparable performance to the base model, indicating no degradation on regular prompts.*

| Models | PartiPrompts-simple |
| --- | --- |
| FLUX | 93.46 |
| SAP | 93.06 |

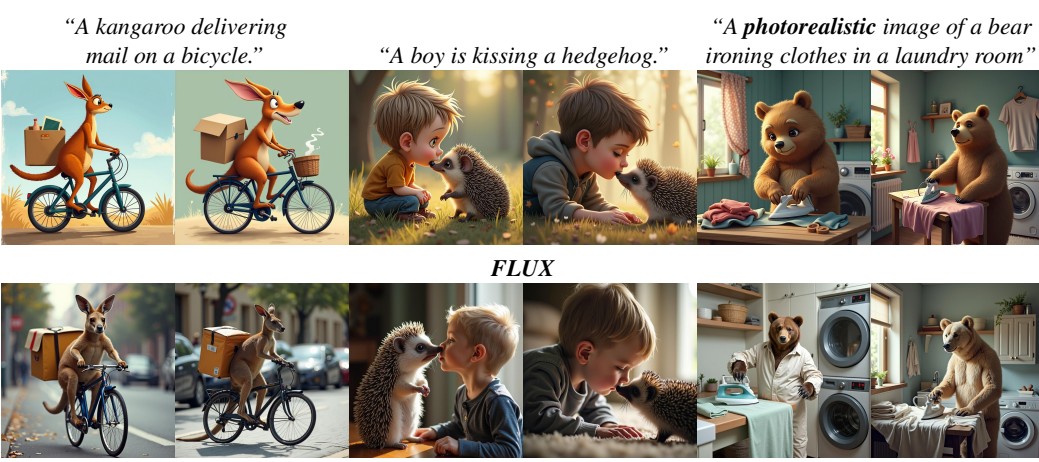

*"A kangaroo delivering mail on a bicycle."*  *"A boy is kissing a hedgehog."*  *"A **photorealistic** image of a bear ironing clothes in a laundry room"*

*FLUX*

*SAP*

Figure 10: *FLUX tends to generate realistic images by default. However, when given unrealistic prompts, it often produces cartoon-like samples, even when explicitly prompted with a "photorealistic" style. In contrast, our method, which gradually resolves such prompts through coherent proxy stages, consistently generates realistic and semantically aligned images.*

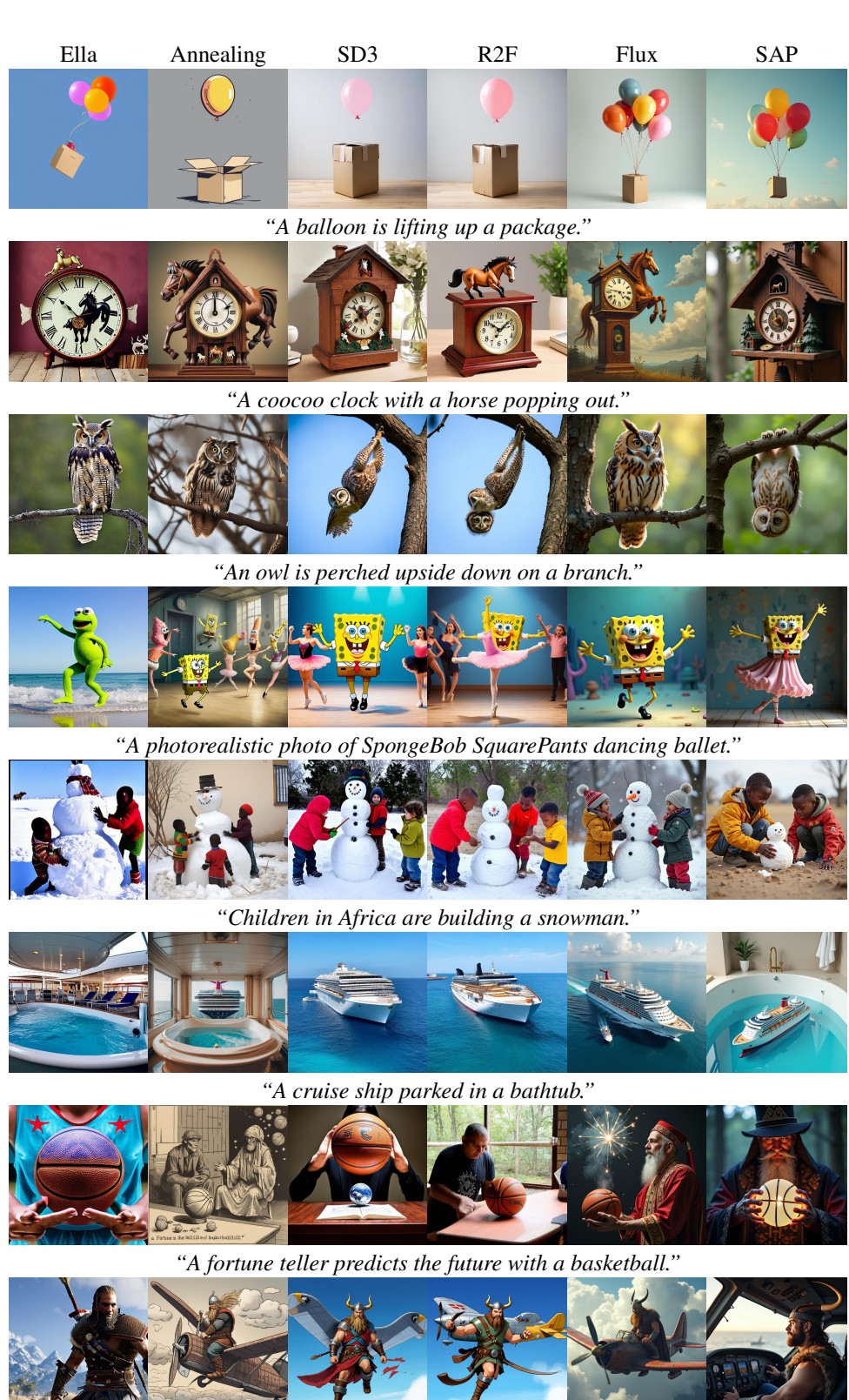

Figure 11: *Qualitative comparison. Our method consistently generates text-aligned images for contextually contradicting prompts.*

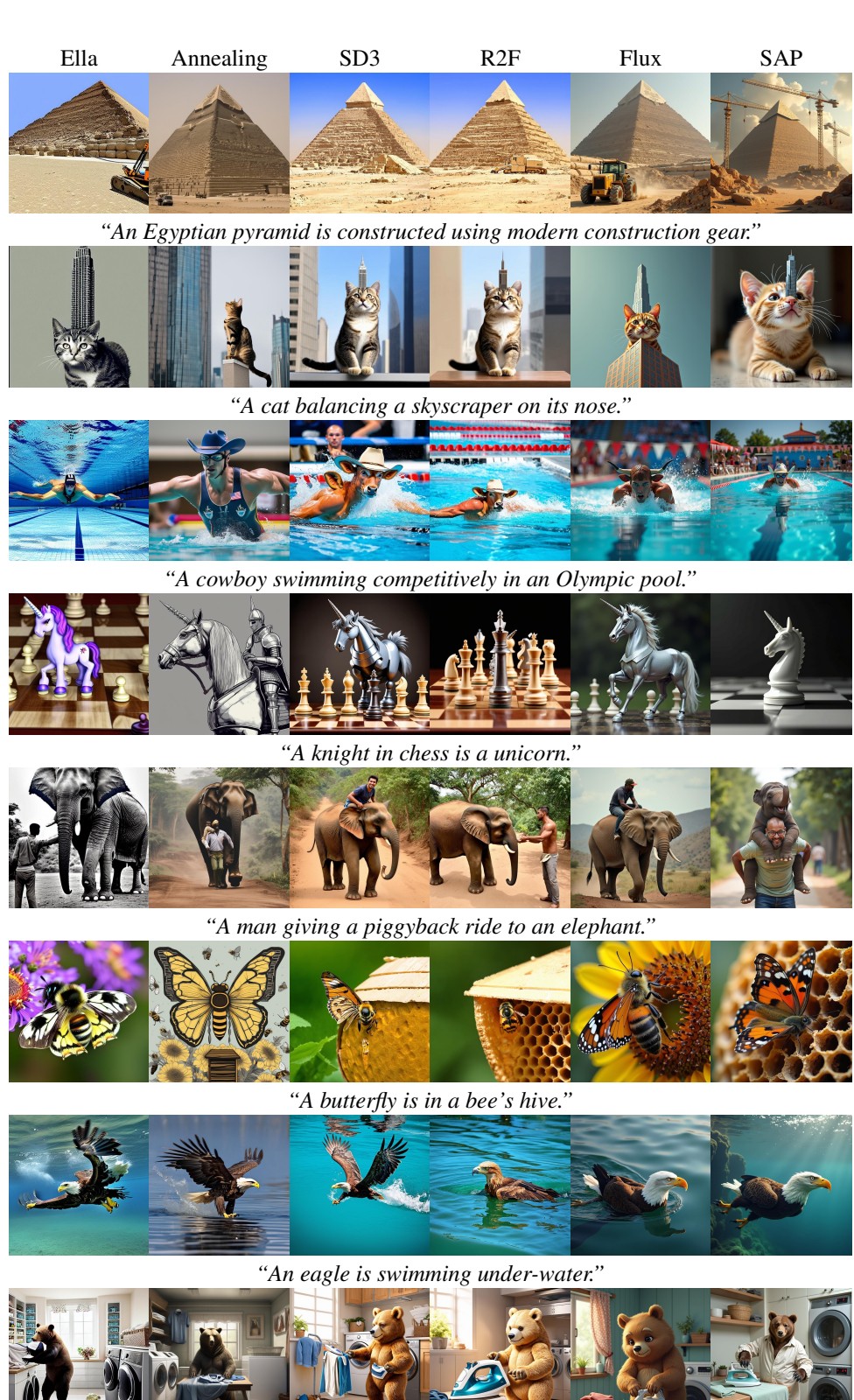

| Ella | Annealing | SD3 | R2F | Flux | SAP |

*"An Egyptian pyramid is constructed using modern construction gear."*

*"A cat balancing a skyscraper on its nose."*

*"A cowboy swimming competitively in an Olympic pool."*

*"A knight in chess is a unicorn."*

*"A man giving a piggyback ride to an elephant."*

*"A butterfly is in a bee's hive."*

*"An eagle is swimming under-water."*

*"A photorealistic image of a bear ironing clothes in a laundry room"*

Figure 12: *Qualitative comparison. Our method consistently generates text-aligned images for contextually contradicting prompts.*

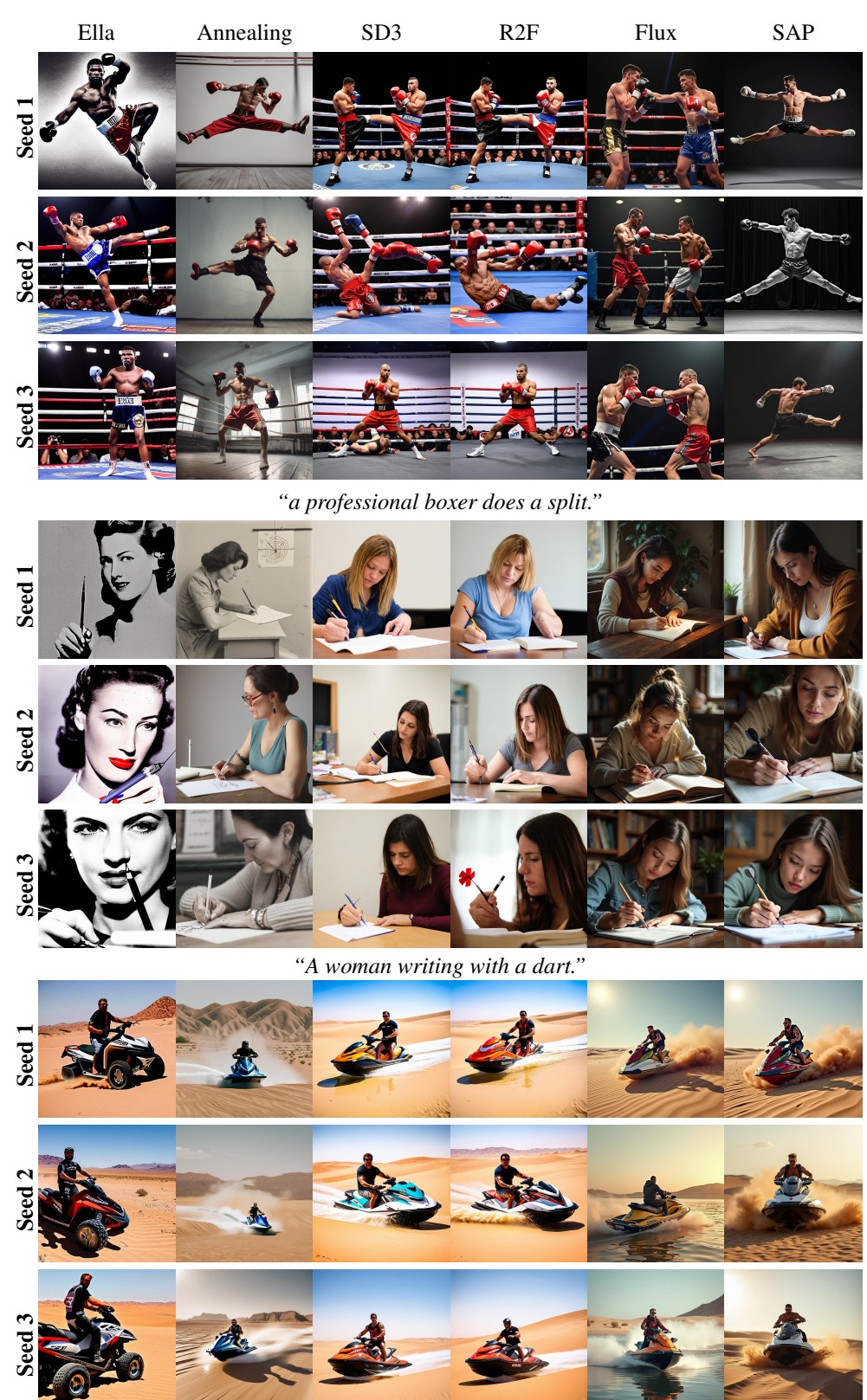

Figure 13: *Qualitative comparison across multiple seeds. Our method consistently generates text-aligned images for contextually contradicting prompts.*

# B   LLM INSTRUCTION FOR PROMPT DECOMPOSITION

Tables 7 and 8 detail the full LLM instruction used for our method's decomposition, along with the corresponding in-context examples. In a single inference pass, our method detects contextual contradictions, generates proxy prompts, and assigns timestep intervals.

Table 9 presents examples of our LLM input prompts, along with the corresponding output explanations and the decomposition into proxy prompts and timestep intervals.

Table 7: *Full LLM prompt instruction SAP, used to decompose prompts by denoising stages.*

**\<System Prompt\>**
You are an expert assistant in time step dependent prompt conditioning for diffusion models.
Your task is to decompose a complex or contextually contradictory prompt into up to **three** intermediate prompts that align with the model's denoising stages — from background layout to object identity to fine detail. Only introduce prompt transitions when needed.

**Diffusion Semantics (Low → High Frequency Progression):**
**Steps 0–2:** Scene layout and dominant color regions (e.g., sky, forest, sand tone)
**Steps 3–6:** Object shape, size, pose, and position
**Steps 7–10:** Object identity, material, and surface type (e.g., glass vs. rubber)
**Steps 11–13+:** Fine features and local details (e.g., tattoos, insects, facial detail)
Since denoising progresses from coarse to fine, it is crucial to stabilize large-scale visual structures (such as body shape, pose, and background) before introducing small or semantically charged elements (such as facial details, objects in hand, or surreal features).

**Substitution Strategy:**
1. Begin with high-level layout (background, geometry).
2. Use **placeholder concepts** if needed to stabilize layout before detailed insertions.
3. Substitutes must match in shape, size, and visual function.
4. Replace placeholders as soon as fidelity permits.
5. Do not maintain substitutions longer than needed.
6. If the prompt is visually coherent, return a **single prompt** with no decomposition.

**Output Format:**
**a. Explanation:** A short sentence explaining why decomposition is needed.
**b. Final dictionary:**

```
{
  "prompts_list": [
    "<prompt1>",
    "<prompt2>",
    "...",
    "<target prompt>"
  ],
  "switch_prompts_steps": [<step1>, <step2>, ...]
}
```
- The length of switch_ prompts_steps should be one less than prompts_list. - Do not include any text outside this structure.

Table 8: *In-context examples SAP, illustrating reasoning steps and prompt decomposition across denoising stages.*

**Input:** A polar bear in a desert
**Output:**
a. Explanation: A polar bear is common in snowy scenes, not deserts. Since no suitable object proxy exists, the prompt starts with the desert alone before introducing the unlikely animal.
b. Final dictionary:

```
{
  "prompts_list": [
    "A desert",
    "A polar bear in a desert"
  ],
  "switch_prompts_steps": [2]
}
```

**Input:** A fox in a nursery
**Output:**
a. Explanation: A fox is uncommon in indoor scenes. Starting with a dog, then a visually similar breed (Shiba Inu), provides a natural proxy before introducing the fox in a childlike setting.
b. Final dictionary:

```
{
  "prompts_list": [
    "A dog in a nursery",
    "A Shiba Inu dog in a nursery",
    "A fox in a baby room"
  ],
  "switch_prompts_steps": [4, 7]
}
```

**Input:** A shoes tree in the meadow
**Output:**
a. Explanation: Treating a "shoes tree" like an "apple tree" allows the model to build structure and object layout using familiar patterns. Introducing apples first grounds the scene in a biologically plausible layout before shifting to the surreal concept of shoes as fruit-like objects.
b. Final dictionary:

```
{
  "prompts_list": [
    "tree in the meadow",
    A tree full of apples, in the meadow",
    "A tree full of shoes in the meadow"
  ],
  "switch_prompts_steps": [4,8]
}
```

**Input:** A red sports car parked on a mountain road
**Output:**
a. Explanation: This is a realistic and visually coherent scene, so no decomposition is needed.
b. Final dictionary:

```
{
  "prompts_list": [
    "A red sports car parked on a mountain road"
  ],
  "switch_prompts_steps": []
}
```

Table 9: *Prompt decompositions of contradictory prompts using SAP.*

| Original Prompt | Explanation | Prompt Sequence | Switch Step(s) |
|---|---|---|---|
| A grown man has a baby's pacifier in his mouth. | The grown man is established first for structure, while the pacifier, being a small but semantically significant detail, is added once the main identity is resolved. | 1. A grown man with a small object in his mouth
2. A grown man has a baby's pacifier in his mouth | 4 |
| A dragon is blowing water. | Dragons are more commonly depicted blowing fire. A proxy of white smoke is visually similar to water mist in texture, stabilizing the emission process before resolving the surreal water emission. | 1. A dragon blowing white smoke
2. A dragon blowing water | 3 |
| A pizza with grape toppings. | Pizza with traditional toppings stabilizes the geometry and color before introducing the visually similar yet unusual grape topping. | 1. A pizza with pepperoni toppings
2. A pizza with grape toppings | 3 |
| A coin floats on the surface of the water. | Coins typically sink in water, not float. Starting with a leaf—an object that naturally floats—ensures that this behavior within the scene is handled correctly before introducing the coin. | 1. A leaf floats on the surface of the water
2. A coin floats on the surface of the water | 4 |
| A cockatoo parrot swimming in the ocean. | Cockatoos are birds and naturally do not swim; starting with a simple bird on water stabilizes position and motion. Progressing to a duck, before introducing the cockatoo parrot, eases the transition into the final surreal visual. | 1. A duck swimming in the ocean
2. A parrot swimming in the ocean
3. A cockatoo parrot swimming in the ocean | 3, 6 |
| Shrek is blue. | Shrek is a distinct character with a recognizable green color. Using a simple blue ogre initially sets the stage for a color change before fully introducing Shrek to ensure visual coherence. | 1. A blue ogre
2. Shrek is blue | 3 |
| A professional boxer does a split. | Professional boxers are typically shown in athletic stances related to fighting, not performing a split. Starting with a gymnast performing a split supports the action, introducing a boxer in similar attire balances identity shift without disrupting the pose. | 1. A gymnast performing a split
2. A boxer performing a split
3. A professional boxer doing a split | 3, 6 |

## C  PROVIDED BENCHMARKS

We describe the construction of *ContraBench* and *Whoops-Hard* in the main text (Section 5). Here, we provide the full lists of prompts for these benchmarks in Table 10 and Table 11, respectively.

Table 10: *ContraBench. A curated benchmark of 40 contradictory prompts for evaluating text-to-image models.*

| ID | Prompt | ID | Prompt |
|---|---|---|---|
| 1 | A professional boxer does a split | 21 | A mosquito pulling a royal carriage through Times Square |
| 2 | A bear performing a handstand in the park | 22 | A grandma is ice skating on the roof |
| 3 | A bodybuilder balancing on point shoes | 23 | A baseball player backswing a yellow ball with a golf club |
| 4 | A chicken is smiling | 24 | A house with a circular door |
| 5 | A cruise ship parked in a bathtub | 25 | A photorealistic image of a bear ironing clothes in a laundry room |
| 6 | A man giving a piggyback ride to an elephant | 26 | A pizza being used as an umbrella in the rain |
| 7 | A zebra climbing a tree | 27 | A cubist lion hiding in a photorealistic jungle |
| 8 | A coffee machine dispensing glitter | 28 | A cowboy swimming competitively in an Olympic pool |
| 9 | A vending machine in a human running posture | 29 | A realistic photo of an elephant wearing slippers |
| 10 | A ballerina aggressively flipping a table | 30 | A computer mouse eating a piece of cheese |
| 11 | A bathtub floating above a desert in a tornado | 31 | A horse taking a selfie with a smartphone |
| 12 | A monkey juggles tiny elephants | 32 | A sheep practicing yoga on a mat |
| 13 | A woman has a marine haircut | 33 | A snake eating a small golden guitar |
| 14 | A tower with two hands | 34 | A soccer field painted on a grain of rice |
| 15 | An archer is shooting flowers with a bow | 35 | A snake with feet |
| 16 | A muscular ferret in the woods | 36 | A woman brushing her teeth with a paintbrush |
| 17 | A barn built atop a skyscraper rooftop | 37 | A horse with a hump |
| 18 | A cat balancing a skyscraper on its nose | 38 | A hyperrealistic unicorn made of origami |
| 19 | A cow grazing on a city rooftop | 39 | A library printed on a butterfly's wings |
| 20 | A fireplace burning inside an igloo | 40 | A photorealistic photo of SpongeBob SquarePants dancing ballet |

Table 11: *Whoops-Hard. A curated subset of 100 challenging prompts from the Whoops!* benchmark.

| ID | Prompt | ID | Prompt |
|---|---|---|---|
| 1 | A bouquet of flowers is upside down in a vase | 51 | A Japanese tea ceremony uses coffee instead of tea |
| 2 | A man is welding without a mask | 52 | A wagon is being pushed from behind by two opposite facing horses |
| 3 | A man eats hamburgers in a baby chair | 53 | The Girl with a Pearl Earring wears a golden hoop earring |
| 4 | A turn right street sign with a left turn arrow | 54 | A chandelier is hanging low to the ground |
| 5 | Goldilocks sleeps with four bears | 55 | The portrait of the Mona Lisa depicts a stern male face |
| 6 | A cake wishes a happy 202nd birthday | 56 | A car with the steering wheel right in the middle of the dashboard |
| 7 | Children are unhappy at Disneyland | 57 | A pagoda sits in front of the Eiffel Tower |
| 8 | An orange carved as a Jack O'Lantern | 58 | A man without protection next to a swarm of bees |
| 9 | A pen is being sharpened in a pencil sharpener | 59 | A kiwi bird in a green bamboo forest |
| 10 | Steve Jobs demonstrating a Microsoft tablet | 60 | The Sphinx is decorated like a sarcophagus outside a Mayan temple |
| 11 | Shrek is blue | 61 | A butterfly is in a bee's hive |
| 12 | A MacBook with a pear logo on it | 62 | A rainbow colored tank |
| 13 | A woman hits an eight ball with a racket | 63 | Movie goers nibble on vegetables instead of popcorn |
| 14 | Vikings ride on public transportation | 64 | A grown man has a baby's pacifier in his mouth |
| 15 | A gift wrapped junked car | 65 | A full pepper shaker turned upside down with nothing coming out |
| 16 | A rainbow is filling the stormy sky at night | 66 | The Tiger King, Joe Exotic, poses with an adult saber-tooth tiger |
| 17 | John Lennon using a MacBook | 67 | A scale is balanced with one side full and the other empty |
| 18 | Michelangelo's David is covered by a fig leaf | 68 | A pizza box is full of sushi |
| 19 | Chuck Norris struggles to lift weights | 69 | A man wearing a dog recovery cone collar while staring at his dog |
| 20 | Paratroopers deploy out of hot air balloons | 70 | A woman's mirror reflection is wearing different clothes |
| 21 | A train on asphalt | 71 | A woman using an umbrella made of fishnet in the rain |
| 22 | Lionel Messi playing tennis | 72 | A field of sunflowers with pink petals |
| 23 | A man jumping into an empty swimming pool | 73 | An eagle swimming under water |
| 24 | An airplane inside a small car garage | 74 | A woman stands in front of a reversed reflection in a mirror |
| 25 | An upside down knife about to slice a tomato | 75 | Stars visible in the sky with a bright afternoon sun |
| 26 | Dirty dishes in a bathroom sink | 76 | A car with an upside down Mercedes-Benz logo |
| 27 | A roulette wheel used as a dart board | 77 | An owl perched upside down on a branch |
| 28 | A smartphone plugged into a typewriter | 78 | A man in a wheelchair ascends steps |
| 29 | A passenger plane parked in a parking lot | 79 | Bach using sound mixing equipment |
| 30 | Guests are laughing at a funeral | 80 | A steam train on a track twisted like a roller coaster |
| 31 | A cat chasing a dog down the street | 81 | Roman centurions fire a cannon |
| 32 | The Statue of Liberty is holding a sword | 82 | A crab with four claws |
| 33 | A Rubik's cube with ten purple squares | 83 | Elon Musk wearing a shirt with a Meta logo |
| 34 | A girl roller skating on an ice rink | 84 | A compass with North South South West points |
| 35 | A butterfly swimming under the ocean | 85 | A glass carafe upside down with contents not pouring |
| 36 | Lightning striking a shack on a sunny day | 86 | Princess Diana standing in front of her grown son, Prince Harry |
| 37 | The Cookie Monster is eating apples | 87 | A children's playground set in the color black |
| 38 | A man is given a purple blood transfusion | 88 | A mug of hot tea with a plastic straw |
| 39 | An unpeeled banana in a blender | 89 | A whole pomegranate inside a corked glass bottle |
| 40 | A square apple | 90 | Belle from Beauty and the Beast about to kiss the Frog Prince |
| 41 | A place setting has two knives | 91 | A person's feet facing opposite directions |
| 42 | A koala in an Asian landscape | 92 | A bowl of cereal in water |
| 43 | A mouse eats a snake | 93 | A boy playing frisbee with a porcelain disk |
| 44 | A field of carrots growing above ground | 94 | A chef prepares a painting |
| 45 | A pregnant woman eating raw salmon | 95 | A dragon blowing water |
| 46 | A tiger staring at zebras in the savanna | 96 | The lip of a pitcher on the same side as the handle |
| 47 | Albert Einstein driving a drag racing car | 97 | Greta Thunberg holding a disposable plastic cup |
| 48 | A soccer player about to kick a bowling ball | 98 | A fortune teller predicting the future with a basketball |
| 49 | An old man riding a unicycle | 99 | A balloon lifting up a package |
| 50 | A hockey player drives a golf ball down the ice | 100 | Bruce Lee in a yellow leotard and tutu practicing ballet |

## D  VLM EVALUATION

We utilize GPT-4o to assess alignment between prompts and their generated images. The instruction prompt provided to the VLM is shown in Table 12.

Table 12: *VLM instruction for evaluation. Used by GPT-4o to score semantic alignment of generated images.*

---

You are an assistant evaluating an image on how well it aligns with the meaning of a given text prompt.

The text prompt is: `"{prompt}"`

---

**PROMPT ALIGNMENT (Semantic Fidelity)**
Evaluate only the *meaning* conveyed by the image — ignore visual artifacts.
Focus on:

- Are the correct objects present and depicted in a way that clearly demonstrates their intended roles and actions from the prompt?
- Does the scene illustrate the intended situation or use-case in a concrete and functional way, rather than through symbolic, metaphorical, or hybrid representation?
- If the described usage or interaction is missing or unclear, alignment should be penalized.
- Focus strictly on the presence, roles, and relationships of the described elements — not on rendering quality.

**Score from 1 to 5:**

- 5: Fully conveys the prompt's meaning with correct elements
- 4: Mostly accurate — main elements are correct, with minor conceptual or contextual issues
- 3: Main subjects are present but important attributes or actions are missing or wrong
- 2: Some relevant components are present, but key elements or intent are significantly misrepresented
- 1: Does not reflect the prompt at all

**Respond using this format:**

```
### ALIGNMENT SCORE: <score>
### ALIGNMENT EXPLANATION: <explanation>
```

---

## E  USE OF LARGE LANGUAGE MODELS (LLMS)

In preparing this paper, we used a large language model (GPT) as an assistive tool for improving grammar, clarity, and wording at the sentence level. In addition, as described in the main text, we employed LLMs in our method and evaluation:

1. Method: As part of our proposed method (Figure 5), an LLM was employed to decompose target prompts into time-dependent proxy prompts (Section 4.2).
2. Benchmark construction: As described in Section 5, ChatGPT was used to generate initial candidate prompts for ContraBench.
3. Evaluation: As explained in Section 5.2, a vision-language model (VLM) was used to assist in evaluating the prompt alignment of the generated outputs.

Beyond these uses, LLMs were not involved in research ideation, experimental design, or the interpretation of results.

