# OpenReview forum: "Image Generation from Contextually-Contradictory Prompts"
_ICLR.cc/2026/Conference — ICLR 2026 Conference Withdrawn Submission_

### Official Review · Reviewer_PnAe · 2025-10-25

**Soundness:** 2
**Presentation:** 3
**Contribution:** 2
**Rating:** 4
**Confidence:** 4

**Summary:**

The authors introduce the concept of contextual contradiction, a failure mode in text-to-image diffusion models where prompts fail not because of direct semantic opposition, but because one concept's learned associations conflict with another concept in the prompt. To solve this, the paper proposes Stage-Aware Prompting (SAP), a training-free framework. SAP is based on the insight that diffusion models generate images in a coarse-to-fine manner (e.g., layout, then shape, then details). The method uses a Large Language Model (LLM) to analyze the target prompt, identify contradictions, and decompose it into a sequence of proxy prompts. These proxies are injected at different timestep intervals of the denoising process to guide the model, first establishing a coherent layout/pose before introducing the contradictory concept. Results from both VLM-based scoring and human user studies show that SAP significantly improves prompt-image alignment compared to baselines.

**Strengths:**

1. The paper clearly articulates contextual contradiction as a specific and challenging problem, distinct from general compositionality failures.
2. Using an LLM to temporally decompose a prompt into a series of proxy prompts that align with the diffusion model's coarse-to-fine generation process is a reasonable, training-free solution.
3. The authors provide a robust and convincing evaluation.

**Weaknesses:**

1. The method's success is critically dependent on the quality of the LLM (GPT-4o) and the 20 manually-crafted in-context examples provided to it. The ablation in Table 3 shows that removing these examples causes a severe performance drop. This indicates the method is less of a general framework and more of a highly-effective prompt engineering strategy, which may not generalize to contradiction types not covered by the 20 examples.
2. The same VLM (GPT-4o) is used as a core component of the method (to generate proxy prompts) and as the primary automated metric for evaluation. Does this introduce a risk of bias in the assessment?
3. The method requires the LLM to output specific timestep intervals (e.g., Steps 0-2, Steps 3-6). These intervals are based on a general heuristic provided in the system prompt, not on an empirical analysis of the specific diffusion models (FLUX, SD3). The paper shows that while the method is robust to minor shifts ($\pm2$ steps), it fails completely if the wrong stage is chosen (Figure 7). This suggests the heuristic must be correct, but its derivation is not well-justified.
4. While the new benchmarks are a strength, they are also small (40 prompts for ContraBench, 100 for Whoops-Hard). This makes it difficult to assess true generalizability.
5. The user study (Table 2) finds that users prefer SAP's outputs for Quality as well as Alignment. The paper does not explain why a prompt-editing method would improve fundamental image quality. This is likely due to resolving semantic conflicts, which prevents the model from generating messy, artifact-ridden hybrid concepts (as seen in baseline failures), but this link should be explicitly stated and analyzed.

**Questions:**

Overall, I like the task proposed in this paper; it is interesting. However, there are still some areas in this paper that need improvement, which are provided in the Weaknesses section. I am open to increasing my score if the authors can effectively address these concerns.

---

### Official Review · Reviewer_Y8yz · 2025-11-01

**Soundness:** 3
**Presentation:** 3
**Contribution:** 2
**Rating:** 2
**Confidence:** 4

**Summary:**

This paper investigates a failure mode in text-to-image diffusion models—contextual contradiction, where concepts in a prompt implicitly negate or conflict due to entangled priors. The authors introduce a stage-aware prompt decomposition pipeline: an LLM analyzes the input prompt, detects contradictions, and emits a sequence of proxy prompts aligned to different denoising stages, guiding sampling toward better semantic faithfulness. Experiments report sizable gains on contradiction-heavy prompts with both automatic metrics and human evaluation.

**Strengths:**

- **Clear figures.** Figures (e.g., Fig. 1, Fig. 5) effectively show the key motivation and the proposed pipeline.
- **Strong results.** The method outperforms baselines by a meaningful margin in reported settings.
- **Evaluation breadth.** The paper includes sufficient evaluation including human study.

**Weaknesses:**

1. **Problem prevalence** It’s unclear how prevalent “contextual contradiction” remains in SOTA systems. I tested gpt-image-1 with the prompt *“Bruce Lee is dressed in a yellow leotard and tutu practicing ballet”* and it performed reasonably well. Many evaluated backbones are ~1 year old. During the period, unified understanding and generation models, like Janus and Show-o surge and they often have great prompt alignment ability. Please evaluate on recent models using your contradiction prompts to quantify the gap and test your method on these models/
2. **Source of the issue: diffusion vs. text encoder.** Several tested backbones rely on relatively weaker text encoders (e.g., CLIP variants for SD). If LLM/LMM-based encoders already resolve many contradictions, the failure may stem more from **language encoding** than the **image generator**. Please:
   - Compare with models that use LLMs/LMMs for prompt parsing (e.g., Qwen-image–style setups).
   - Analyze whether your gains persist when a stronger language encoder is used.
3. **Novelty positioning.** There is a bunch of work using LLMs to ground/composite prompts or alter conditioning across steps. Also, How is “contextual contradiction” distinct from known **bias/entanglement** issues (e.g., color or gender bias) beyond terminology.

**Questions:**

1. **Interval assignment.** Are stage intervals decided solely by the LLM? How do you ensure LLMs have a good understanding of the denoising process? Can you try a VLM that periodically inspects partially denoised images to decide when to advance to the next stage? But it should suffer from higher compute cost.
2. **Benchmark construction.** You construct **Whoops-Hard** from **Whoops**. How are prompts selected? What criteria define “hard”?
3. **Cost/latency & robustness.** Your work uses llm as a component, can you show the cost of it?

---

### Official Review · Reviewer_nDwe · 2025-11-01

**Soundness:** 2
**Presentation:** 3
**Contribution:** 3
**Rating:** 2
**Confidence:** 5

**Summary:**

This paper generates images for cases where contextual contradictions prevail. For example, a lion doing a  headstand would generally generate a human doing headstand with models like FLUX. To achieve this, based on the widely known and studied observation that diffusion models generate images in coarse to fine fashion, with low-level details in the early timesteps and finer details in the last timesteps,
the paper leverages multi-stage prompting strategy. LLM is provided with instruction that given a prompt as input, it decomposes it into a series of prompts which are then input to the diffusion model for generation
The method is compared on Whoops dataset for contradictory generation against FLUX, SD3, annealing, R2F using LLM as a judge.

**Strengths:**

The paper proposes a training free approach for generating contextually contradictory prompts.
The proposed approach outperforms FLUX, SD3.0, R2F, Ella, Annealing Guidance in both alignment and visual quality across multiple dataset.
The qualitative examples also show the effectiveness of the approach.

**Weaknesses:**

1. The details on the attention mechanism in Figure 3 are not clear.
2. Construction of in-context examples: The prompts considered have limited diversity with variations in the object and the background
3. The approach considers limited variations in the visual style with respect to which the modifications are performed. For example, a single instance of multi-object scenario will be challenging
4. Comparison to approaches or negative prompting techniques: The paper focuses on progressive addition of concepts.  Negative prompting techniques can be considered.
5. Evaluation is based solely on LLM as a judge. If same LLM used to create prompts and evaluate it can introduce bias from its own prompting strategy.

**Questions:**

1. Details on the model used for generation and the attention visualization is not clear. From which layer are they taken or averaged? Which model is used?
2. How does the approach generalize to scenarios with multiple foreground concepts or with text-centric visual information?
3. How would the approach benefit from negative prompting?
4. Can the authors comment on the evaluation strategy and its robustness.  Evaluating CLIP similarity can be considered?

**Details Of Ethics Concerns:**

The paper generates content which contradicts the correlations which at times are desirable. Its implications on broader Generative AI applications need to be checked.

---

### Official Review · Reviewer_j113 · 2025-11-03

**Soundness:** 3
**Presentation:** 3
**Contribution:** 3
**Rating:** 6
**Confidence:** 3

**Summary:**

This paper tackles a common issue in text-to-image models: they often fail when prompts contain “contextually contradictory” concepts (e.g., a butterfly in a bee’s hive). The authors propose a method called Stage-Aware Prompting (SAP), which uses an LLM to break down the original prompt into simpler, non-conflicting proxy prompts. These are then used at different stages of the image generation process to guide the model toward more accurate results.

**Strengths:**

The biggest strength is the elegant simplicity of the approach. It doesn't require retraining the base model, making it easy to apply to existing systems like FLUX or SD3. I was particularly impressed by the use of an LLM as a "planner" to resolve semantic conflicts—it feels like a natural and powerful fit. The results are convincing, showing clear improvements on challenging benchmarks where other methods produce nonsensical hybrids. It's also a major plus that the method doesn't degrade performance on standard, non-contradictory prompts.

**Weaknesses:**

The approach's reliance on the LLM is also its main weakness. The quality of the final image is now dependent on the LLM's ability to correctly diagnose the contradiction and generate sensible proxies, which might not always be robust. Additionally, the method can only work around the base model's limitations; it won't fix fundamental issues like incorrect object counts or strange anatomy if the diffusion model itself struggles with them. The process for defining the timestep intervals for switching prompts also feels a bit heuristic and could benefit from a more principled design.

**Questions:**

How does the method perform with less capable, open-source LLMs (like LLaMA)? The paper briefly tests this, but is the performance drop acceptable for practical use?

Could this framework be extended beyond "contextual" contradictions to handle more direct logical or physical impossibilities (e.g., "a square apple")?

The LLM and diffusion model are separate. Have the authors considered a more tightly integrated design, where a VLM's understanding could directly influence the denoising process in real-time?

---

### Note · Authors · 2025-11-13

I have read and agree with the venue's withdrawal policy on behalf of myself and my co-authors.